# Vertical Wind and Drop Size Distribution Retrieval with the CloudCube G-band Doppler Radar

Nitika Yadlapalli Yurk[1], Matt D. Lebsock[1], Juan M. Socuellamos[1], Raquel Rodriguez Monje[1], Ken B. Cooper[1], and Pavlos Kollias[2]

[1]Jet Propulsion Laboratory, California Institute of Technology, Pasadena, California, USA
[2]Division of Atmospheric Sciences, Stony Brook University, Stony Brook, NY, USA

**Correspondence:** Matt D. Lebsock (matthew.d.lebsock@jpl.nasa.gov)

**Abstract.** Macrophysical properties of clouds are influenced by underlying microphysical processes. In practice, there is often an observational gap in bridging the two. For example, our current understanding of aerosol-cloud interaction and cloud-climate feedback is hindered by a lack of robust measurements of the distribution of drop sizes within clouds, especially for the smallest drop sizes. Doppler radar measurements have proven useful in estimating rainfall drop size distributions (DSDs) but face an intermediate challenge of requiring a correction for the presence of vertical air motion. Recent advances in millimeter wave technology have made radar measurements at ever smaller wavelengths possible, allowing for analysis of particle size dependent scattering effects to back out estimates of vertical winds and thereby DSDs. This work demonstrates a method of deriving range-resolved DSDs using Doppler spectra at 238 GHz measured by the CloudCube ground-based G-band atmospheric Doppler radar. The observations utilized are of marine boundary layer clouds during March and April 2023 in La Jolla, CA, USA, taken as part of CloudCube's participation in the Eastern Pacific Cloud Aerosol Precipitation Experiment (EPCAPE) campaign. This method first identifies notches in the velocity spectra and compares them to the theoretical notch velocities predicted by size dependent backscattering and terminal velocity models to estimate the range-dependent vertical wind. After removing the vertical wind, binned DSDs are retrieved from the zero-wind spectrum. Bulk properties of the precipitation are then derived including the number concentration, liquid water content, characteristic drop size, and precipitation rate. For the case study presented here, calculated bulk properties are found to be relatively invariant to the forward model assumptions made in the estimation of the full DSD retrieval. Validation of this method on larger volumes of data would make such retrievals useful tools in assessing physical models of drizzle

## 1 Introduction

Marine boundary layer clouds represent the largest physical source of uncertainty in projections of climate sensitivity (Zelinka et al., 2020) and are central to understanding the radiative forcing of aerosol-cloud interactions (Bellouin et al., 2020). A consistent finding is a relationship between the occurrence of precipitation and the mesoscale organization of low clouds (Abel et al., 2017; Yamaguchi et al., 2017; Smalley et al., 2022), where a transition from closed cell clouds to open cell clouds is associated with precipitation onset. Therefore, these cloud transitions are critical in constraining both aerosol-cloud interaction

and cloud-climate feedbacks. A current dilemma in climate projection is the fact that the accuracy of future projections are limited by a negative correlation between aerosol-cloud interactions and cloud-climate feedback (Gettelman et al., 2024). This anti-correlation has been clearly linked to climate model representation of the precipitation formation process (Suzuki et al., 2013).

Accurate measurements of drizzle and light rain are essential to improve understanding of the microphysical processes in boundary layer clouds and constrain both the aerosol-cloud interactions and the cloud-climate feedback. Understanding how the size distribution of drizzle drops evolves in both time and space can provide insight into coalescence, breakup and evaporation processes that shape cloud macrophysical properties. Current methods of directly measuring DSDs include using either ground-based or airborne disdrometers, devices which directly measure drop sizes. Ground-based disdrometers only measure drops which fall all the way to the surface and have a limited capability to observe the smallest precipitation drops (Wang and Bartholomew, 2023). Airborne measurements are more likely to capture data at several elevations, however these measurements are sparse.

Continuous observations from remote sensing measurements are necessary to fill the gaps in in-situ sampling. Radar is the optimal tool for remote observations of drizzle and light rain from ground-based or airborne platforms. The most straightforward method to derive drizzle parameters is by assuming a reflectivity drizzle-rate (Z-R) relationship (Comstock et al., 2004). However, in practice Z-R relationships have primarily been used operationally for satellite cloud radar observations where reliable Doppler observations are not available (Lebsock and L'Ecuyer, 2011; Mroz et al., 2023). The widespread proliferation of mm-wave Doppler cloud radars enabled a new class of retrieval of drizzle and light rain that combine radar reflectivity and higher Doppler moments. For example, Frisch et al. (1995) combine Ka-band Doppler spectral moments with an assumption of zero vertical wind and an assumed drop size distribution shape to derive the vertical profile of drizzle parameters. O'Connor et al. (2005) advanced on this approach by combining W-band Doppler moments with lidar backscatter and a method to correct for turbulent broadening. This multi-sensor approach has subsequently been used to make novel observations of drizzle in stratocumulus clouds (Ghate and Cadeddu, 2019). Galloway et al. (1999) invert an airborne W-band Doppler spectrum to derive a binned drizzle DSD without assuming a DSD shape while retaining the zero mean wind assumption.

One common shortfall of the Doppler-based methods mentioned above is the difficulty in accounting for the effect of the vertical air motion on the mean Doppler. In this respect, Mie scattering in mm-wave radars can be useful in constraining the vertical air motion when the size of drops is similar to the observing wavelength of a radar system. Specifically, the backscattering efficiency at a particular observing wavelength contains several peaks and valleys as a function of drop radius, as seen in Fig. 1. Lhermitte (1987) first proposed the technique of using full W-band Doppler spectra that show similarly oscillatory shapes along with information about the theoretical backscattering efficiency in each velocity bin to simultaneously retrieve information about the vertical air velocity and the rain DSD. The difference between the theoretical and observed locations of any minima seen in the Doppler spectrum would yield information about vertical air motion in the scene while the relative heights of any maxima seen in the spectrum would yield information about the DSD of the scattering particles. Kollias et al. (2002) demonstrated vertical air motion retrievals derived from W-band Doppler spectrum observations of stratiform precipitation. Giangrande et al. (2010) first demonstrated the full utility of this technique to analyze W-band Doppler spectra.

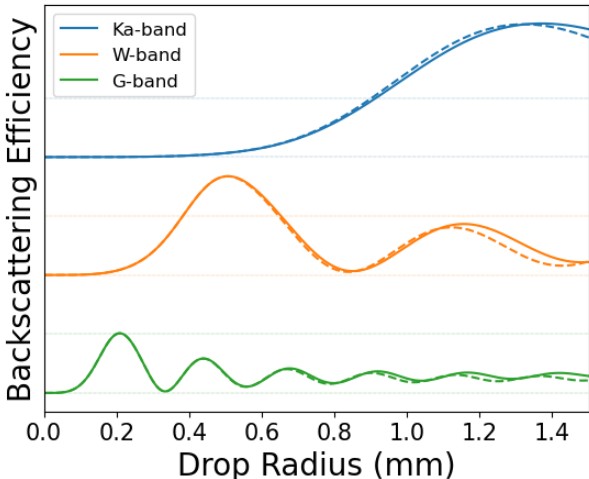

**Figure 1.** Backscattering efficiency as a function of particle radius for three different radar bands. The solid lines represent backscattering calculated with the T-matrix method, which assumes some oblateness of the drops. The dotted lines represent Mie backscattering, which assumes the drops are spherical. The horizontal dotted lines represents lines of efficiency equal to 0 and 1 for each band. Calculations are described in Sect. 2.3

This work retrieved measurements of both vertical winds as well as best-fit parameters to a Marshall-Palmer log-linear DSD; however, useful spectra can only be captured at W-band for storms with large droplets greater than ~0.8 mm. As seen in Fig. 1, the first backscattering minimum occurs at a larger drop radius for lower observing frequencies. Thus, to be sensitive to measurements of drops as small as drizzle (smaller than a diameter of 0.5 mm), it is necessary to make Doppler spectrum measurements at a higher frequency. In particular for G-band, the first minimum is located at a small enough radius that its location is insensitive to the parametrization of the drop aspect ratio, permitting high accuracy quantification of the vertical air motion in all but the lightest liquid phase precipitation. The utility of G-band observations was theorized by Battaglia et al. (2014) and first demonstrated by Courtier et al. (2024) in the addition of G-band Doppler spectra to a multi-frequency DSD retrieval.

This work explores the capability of making retrievals based only on G-band spectra to profile liquid phase precipitation DSDs in marine boundary layer clouds. This class of precipitation is ideally suited for G-band for two reasons: (1) the preponderance of small drops means that the Mie resonance (or notch) will frequently not be observed in W-band spectra but will be observed at G-band, and (2) the liquid water content is small and thus the hydrometeor attenuation tends to be small. The latter fact means that attenuation correction can be performed without incurring the large errors common in rainfall retrievals at frequencies that are used to observe more highly attenuated conditions (e.g. Ka-band, W-band; Hitschfeld and Bordan, 1954). To demonstrate these capabilities, the paper uses the first operational data from a deployment of a newly developed G-band Doppler cloud radar to a large field deployment at a coastal site with frequent marine boundary layer clouds and validates the results against ancillary observations.

## 2 Instrument and Data Overview

### 2.1 CloudCube Instrument

CloudCube is a modular triple-frequency (Ka-band, W-band, and G-band) atmospheric radar instrument developed at the Jet
Propulsion Laboratory. Its use of both a fully solid state design as well as direct up-/down-conversion between baseband and
RF allows it to have a uniquely compact architecture, ideal for deployment in the field. This paper focuses specifically on
the G-band) channel, currently the only one of CloudCube's with full Doppler spectral resolution. The observing frequency,
238.8 GHz, was strategically chosen to take advantage of an intersection between an allowed frequency allocation and a trough
in the atmospheric absorption curve. It also lies close to the limits at which transmit sources of sufficient power are available.
A summary of the instrument parameters are shown in Table 1, and more detail on the instrument can be found in Socuellamos
et al. (2024a).

### 2.2 EPCAPE Campaign

The data presented in this paper were collected as part of CloudCube's participation in the Eastern Pacific Cloud Aerosol
Precipitation Experiment (EPCAPE), a US Department of Energy's (DOE) Atmospheric Radiation Measurement (ARM) cam-
paign Russell et al. (2021). The main goal of EPCAPE was to better understand marine stratocumulus clouds and their effect
on Earth's radiation budget. CloudCube measured cloudy and lightly raining cumulus and drizzling stratocumulus over several
days during March and April 2023 from atop Scripps Pier in La Jolla, CA, stationed adjacent to the ARM Mobile Facility
(AMF). Data from all three bands of CloudCube were saved during this time. Details of the post-processing for the data can be
found at Socuellamos et al. (2024c), and the datasets are made publicly available in Socuellamos et al. (2023). The majority of
the precipitation events during this deployment period were not observed by CloudCube because at that time the instruments
did not have radomes and had to be covered during periods of surface precipitation to protect the radars.

This paper also uses data taken from several ARM instruments to both aid and supplement the presented analysis. Notably,
our retrievals rely on temperature, pressure, and humidity profiles collected by radiosondes (Holdridge, 2020) for determining
the correct values of particle backscattering efficiencies, fall speeds, and gaseous attenuation. The retrievals are validated with
the ARM Ka-band radar (KaZR; Widener et al., 2012, see radar parameters in Table 1) and the 2-D video disdrometer (VDIS;
Bartholomew, 2020) instruments.

### 2.3 Scattering Properties

The single scattering properties of liquid precipitation drops are calculated with the T-Matrix method (Mishchenko and Travis,
1998), using the Python wrapper of Leinonen (2014). The aspect ratio of drops is modeled using the equation $\frac{b}{a} = 1.055 -
0.0653D$ where $D$ is the drop diameter in mm, valid in the range 1.5–8 mm (Thurai and Bringi, 2005), and $b/a$ is the axis
ratio of the spheroids. The aspect ratio is equal to one for the smallest drops (smaller than a diameter of 0.84 mm) for which
this formula produces aspect ratios larger than unity. A look-up-table is created with the drop single scattering properties in

|  | KAZR | CloudCube G-band |
|---|---|---|
| **Frequency (GHz)** | 34.89 | 238.8 |
| **Transmission type** | Pulsed | FMCW |
| **Pulse width ($\mu s$)** | 0.3 | 40 |
| **Pulse repetition interval (ms)** | 0.27 | 0.042 |
| **Peak transmit power (W)** | 100 | 0.24 |
| **Antenna beamwidth (deg)** | 0.19 | 0.35 |
| **Range resolution (m)** | 30 | 10 |
| **Unambiguous range (km)** | 40 | 6.3 |
| **Velocity resolution (ms$^{-1}$)** | 0.02 | 0.06 |
| **Nyquist velocity (ms$^{-1}$)** | $\pm 7.97$ | $\pm 7.5$ |
| **Time resolution (s)** | 4 | 0.4 |

**Table 1.** Summary of KAZR and CloudCube G-band radar specifications at EPCAPE. FMCW = frequency modulated continuous wave

one micron increments in radius and 1 K increments in temperature. The temperature dependent refractive index is taken from Warren (1984). At the CloudCube observing frequency (238 GHz), the first minimum is located at a drop radius of 0.33 mm.

## 3 Data Filtering and Minima Finding

CloudCube collected around 51 hours of data over 13 separate days during its deployment. However, only a few instances spread over two days of this dataset contained spectra resolving at least one backscattering minimum for at least 0.5 continuous km and 100 continuous seconds to perform robust retrievals. These values are chosen arbitrarily - a future work may investigate selecting these thresholds adaptively. The spectrum in Fig. 2b, referred to hereafter as "the example spectrum," will be used to demonstrate our methods for the remainder of this paper.

To provide context for this example spectrum, Fig. 2a shows the G-band reflectivity curtain and marks the time at which the example spectrum was collected. Also shown is the cloud base height as measured by the ARM infrared laser ceilometer (Morris, 2016). Fig. 2c shows a characteristic Doppler spectrum of a precipitating post frontal shallow cumulus ice-phase cloud. This cloud happens to have a cloud base near the melting level which can be very clearly observed in Fig. 2b as the region near an altitude of 1.4 km below which the narrow Doppler spectrum with small Doppler velocity rapidly broadens as large ice crystals melt into falling rain drops with significantly increased Doppler velocities. Note also how the melting layer appears as a weak bright band in panel (a) and a rapid change in the mean Doppler velocity in panel (b). In the liquid precipitation below the melting layer there are multiple resonances in the Doppler spectra corresponding to the Mie notches that can be exploited to find the vertical air motion. As the scattering properties of liquid drops as well as the resulting Doppler spectra are relatively

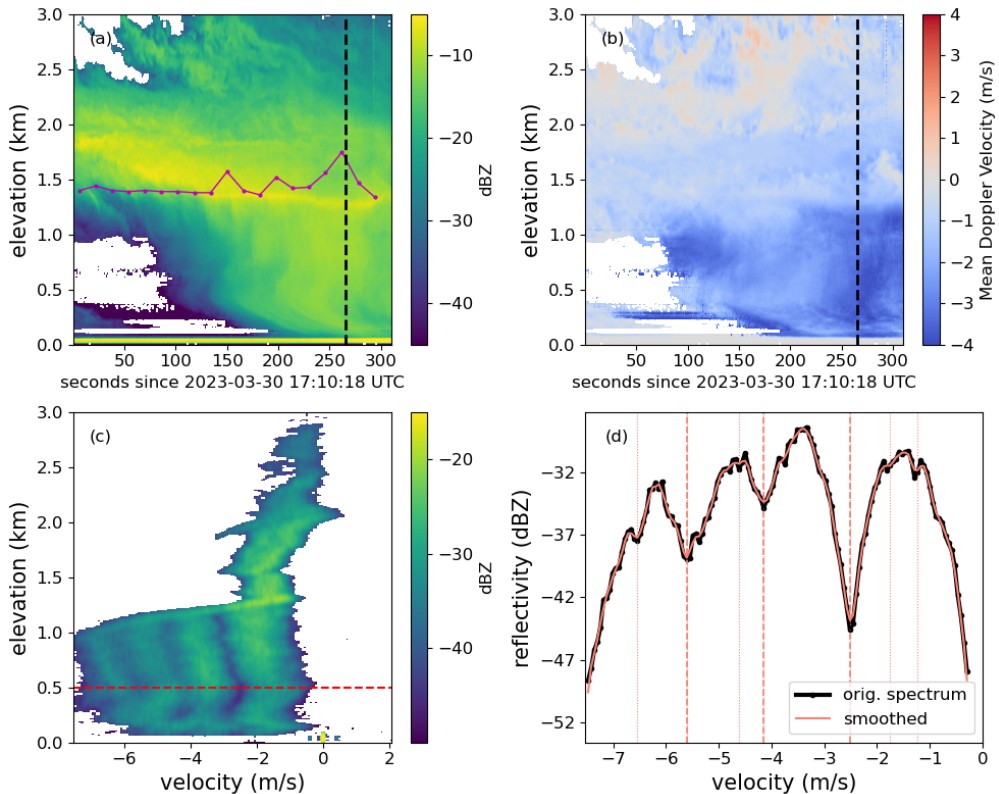

**Figure 2. (a)**: Reflectivity curtain for several minutes of G-band radar data collected on 30 March 2023. The pink line shows the location of the cloud base, as measured by the ARM laser ceilometer. **(b)**: Mean Doppler velocity corresponding to the reflectivity curtain in (a). **(c)**: A full Doppler spectrum from data collected during the time marked by the black dotted line in (a). The dashed red line shows the elevation for the example 1-D spectrum on the right. **(d)**: The black line represents the original measured spectrum whereas the pink line shows the Wiener smoothed spectrum. The thicker dashed vertical lines represent minima in the spectrum that are considered to be truly correlated with minima in the backscattering function whereas the thin dashed lines represent other candidate minima the `find_peaks` function found without enforcing any checks.

straightforward to calculate compared to the complexity of modeling ice, this work limits itself to analyzing regions where only liquid water is present. An investigation of ice retrievals is saved for a future work.

To begin, the spectra are smoothed in the range direction to filter out the smallest scale of vertical turbulence, improving the signal to noise in the minima detection. The spectra originally have a range resolution of 10 m, and a 1-D Gaussian blurring kernel is used to smooth to an effective range resolution of 50 m. Following the smoothing step, each 1-D velocity power spectrum is analyzed to identify any minima.

To discern true backscattering minima from Rayleigh-distributed noise fluctuations, we search for minima using a smoothed version of each 1-D spectrum. The spectrum is smoothed using a Wiener filter and use the difference between the smoothed and

original spectra to estimate the standard deviation of the noise fluctuations. Next we utilize the Python function `scipy.find_peaks` (Virtanen et al., 2020), which identifies peaks as any points where its two neighbors are of a lower value – using sign-inverted spectra allows the minima to show up as peaks. To separate true minima from spurious ones, a few criteria are imposed. We mandate that for true detections, the depth of the minima (measured as the peak to trough distance between the minimum and the nearest peak) must be at least five times the standard deviation of the noise fluctuations. Based on the spacing between successive minima seen in the T-matrix backscattering calculation, it is enforced that the spacing between minima found in the spectra must be separated by at least $1.15\,\mathrm{m\,s^{-1}}$. For sets of minima found with smaller spacings, minima with lower signal to noise ratios (SNRs) are preferentially filtered out. Our final check is to ensure that we are not erroneously finding points in the noise floor of the spectrum as minima by ensuring that for true detections there are continuous data points for at least $1\,\mathrm{m\,s^{-1}}$ on each side of the minimum. A demonstration of this minima retrieval for a single elevation is shown in Fig. 2c.

After minima at all elevations have been identified in a Doppler spectrum profile from a single time, they are classified based on which minimum in the backscattering function they correlate to. The simplest method of doing this would be to draw boundaries of fixed width in velocity space and assume all points within each section are correlated to the same backscattering notch. However, this requires an a priori assumption of a mean vertical wind value with relatively low variance across elevation. To mitigate the risk of poor retrievals resulting from incorrect initial assumptions, a Gaussian mixture model (GMM) is used to cluster points associated with the same backscattering notch. This method assumes that all points in a given data set are drawn from one of $N$ multivariate Gaussian distributions, each with their own means and covariance matrices. We utilize the Python implementation of GMM in `sklearn.mixture.GaussianMixture` (Pedregosa et al., 2011). The algorithm initially looks for data points in the $1\,\mathrm{m\,s^{-1}}$ regions around the locations of first three theoretical backscattering minima for $T = 270\,\mathrm{K}$. The number of components for the GMM to sort data into is determined by how many of these initial regions have data within them. For example, Fig. 3a, shows a spectrum with these divisions overlaid. Since data are present in all three of the divisions, a GMM with three components is used. The initial locations of the three Gaussian components are decided using the mean elevation and velocity of the data within each division. The shape of each Gaussian is initialized by a covariance matrix derived from the standard deviations of the elevations and velocities of the data. An example of these initial guesses is also shown in Fig. 3a.

The final components are fit by adjusting the parameters of the Gaussians until the likelihood of all points being drawn from one of the distributions is maximized. The final classifications are adjusted if necessary by ensuring that each height only contains one point from each distribution. Any outlier points that have anomolously low likelihoods are masked. An example of the final distributions can be seen in Fig. 3b. Note the cyan point to the far left of the spectrum that was identified as a minimum but is masked as an outlier due to being many standard deviations away from any of the three Gaussian components. Within each cluster, any remainaing outlier points are identified using the `LocalOutlierFactor` (LOF) method in `sklearn`, which measures the local densities of points and identifies any points that have anomalously low local densities. In this example case, the LOF method identified the cyan point near $250\,\mathrm{m}$, close to the purple points, as being an outlier point. It however, did not identify, the two yellow points that seem, by eye, out of distribution. Tuning this method exactly can be a challenge.

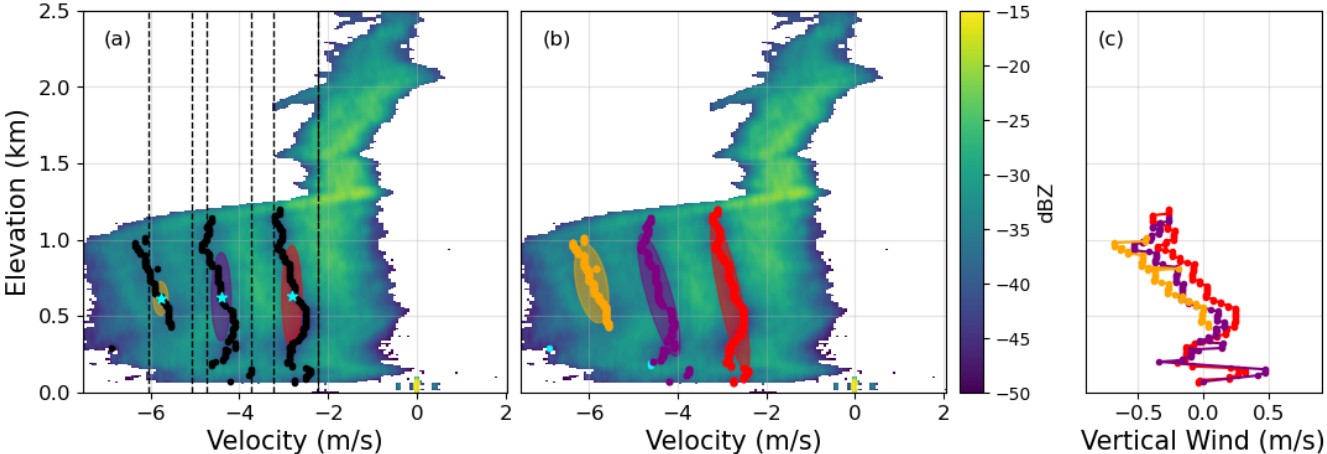

**Figure 3. (a)**: 2-D Doppler spectrum with all identified minima overlaid. The black dashed lines represent the boundaries used to define the initialization of the classifier. The blue stars represent the means of the points within the boundaries and the colored ellipses represent the initial covariances in both the height and velocity directions. **(b)**: 2-D Doppler spectrum with all identified minima classified according to their correlation to notches in the backscattering function. The cyan points are minima that were filtered out as being erroneous and the ellipses show the probability spaces of the Gaussian mixture model components. **(c)**: vertical winds calculated from the classified minima.

## 4 Vertical Wind Retrieval

After classifying the minima, the vertical winds are retrieved from their measured locations. The Gaussian component with the mean velocity closest to zero is assumed to contain the points that correspond to the first backscattering minimum. For each elevation, the temperature measured by the radiosonde is used to select the correct temperature-dependent backscattering efficiency function. As the backscattering efficiency is a function of diameter and our spectrum power is a function of velocity, drop diameters are transformed to drop velocity by assuming that all drops are falling at terminal velocity. For drop diameters greater than $100\,\mu$m, we linearly interpolate between the data points presented in Gunn and Kinzer (1949) to calculate terminal velocity. For smaller drops, we use Stokes law, $v_t = \frac{1}{4}kD^2$ ($k = 1.19 \times 10^8$ is a constant, $D$ in m, $v_t$ is terminal velocity in m s$^{-1}$). The effects of air density are corrected for by multiplying the terminal velocity by a correction factor, $C = (\rho_0/\rho)^m$ where $\rho_0 = 1.204\,\text{kg m}^{-3}$ (density for standard temperature and pressure) and $m = 0.375 + (2.5 \times 10^{-5})D$ (Beard, 1985). The air density, $\rho$, as a function of elevation is calculated using the temperature and pressure values measured by radiosondes. Once backscattering efficiency is transformed to be a function of velocity, the measured minimum value is subtracted from the theoretical value to retrieve vertical wind as a function of height: $v_{\text{wind}}(h) = v_{\text{meas}}(h) - v_{\text{theo}}(h)$. An example of this retrieval using each of the minima is shown in Fig. 3c. The colors of the vertical wind curves correspond to which color of minima which are used to derive the wind speeds.

There are small inconsistencies between wind speeds at the same height calculated from different minima. It is difficult to determine the exact cause of this inconsistency, but is it likely due to a combination of uncertainty in the Gunn-Kinzer terminal

velocity relationship and the drop obliquity parametrization are the largest contributors to this discrepancy. For a monotonically varying DSD, the details of the DSD should not affect the locations of the minima. For the rest of the analyses presented in this paper, only the vertical winds derived from the first minimum are considered (corresponding to the red points in Fig. 3). Recall that the location of the first minimum can be assumed to be insensitive to the drop obliquity parametrization.

## 5 Drop Size Distribution Retrieval

As described in Kollias et al. (2011), the measured Doppler spectra can be described by the equation

$$S(v + v_{wind})_{obs} = (A + \epsilon_a)[S(v)_Q * g(\sigma_{v,turb})] + \epsilon_s \tag{1}$$

where $v$ is the true particle velocity, $v_{wind}$ is the vertical wind speed, $A$ is attenuation, $\epsilon_a$ is the attenuation error, $S(v)_Q$ is the quiet-air spectrum (no turbulence), $g(\sigma_{v,turb})$ is the convolution kernel that describes spectral broadening due to turbulence, and $\epsilon_s$ represents error in the measured spectral power. Retrieval of the quiet-air spectrum from the measured spectrum would enable measurement the drop size distribution as a function of drop radius, $N(r)$, in units of $m^{-3}\,m^{-1}$, using the relationship

$$S(v)_Q = \frac{\lambda^4}{\pi^5 |K(\lambda)|^2} \sigma_{bck}(r) N(r) \frac{dr}{dv_t} \tag{2}$$

where $\lambda$ is the observing wavelength in mm, $|K(\lambda)|^2$ is the dielectric factor at the observing wavelength, and $\sigma_{bck}(r)$ is the backscattering cross section in $mm^2$ as a function of particle size. $S(v)_Q$ has units of $mm^6 m^{-3} (m\,s^{-1})^{-1}$. As the CloudCube spectra are saved in units of dB$Z$, the spectra are transformed to linear units using the relations dB$Z = 10\log_{10}(Z/Z_0)$, $Z_0 = 1\,mm^6 m^{-3}$, and $Z = S(v)\,dv$.

The radius-resolution of the DSD retrieval is defined by the velocity resolution of CloudCube (0.06 ms$^{-1}$) and varies according to $\frac{dr}{dv_t}$. The radius-resolution varies greatly as a function of drop radius. For radii less than 50 $\mu$m, the resolution starts off coarse as $\frac{dr}{dv_t}$ is larger. This value decreases initially as the fall speed of the smallest drops is given by Stoke's law. The resolution value increases with larger drop sizes. Plots of the terminal velocity relationships as well as the radius resolution at standard temperature and pressure are shown in Fig. 4. The variations seem in the radius resolution plot are likely due to measurement errors of the Gunn-Kinzer points.

### 5.1 Turbulence-free Assumption

To begin with the simplest retrieval scheme, we first assume that spectra were captured in a turbulence-free environment. Ignoring for now attenuation error as well, the measured spectrum can then be modeled simply as

$$S(v + v_{wind})_{obs} = AS(v)_Q + \epsilon_s. \tag{3}$$

With this simplification, once attenuation is corrected for, the above equation can be inverted to solve for $N(r)$. Calculation of the spectrum error is carried out according to the analysis presented in the appendix of Hogan et al. (2005). Based on CloudCube's observing wavelength and the scale of wind speeds measured both directly by the sondes and calculated from the

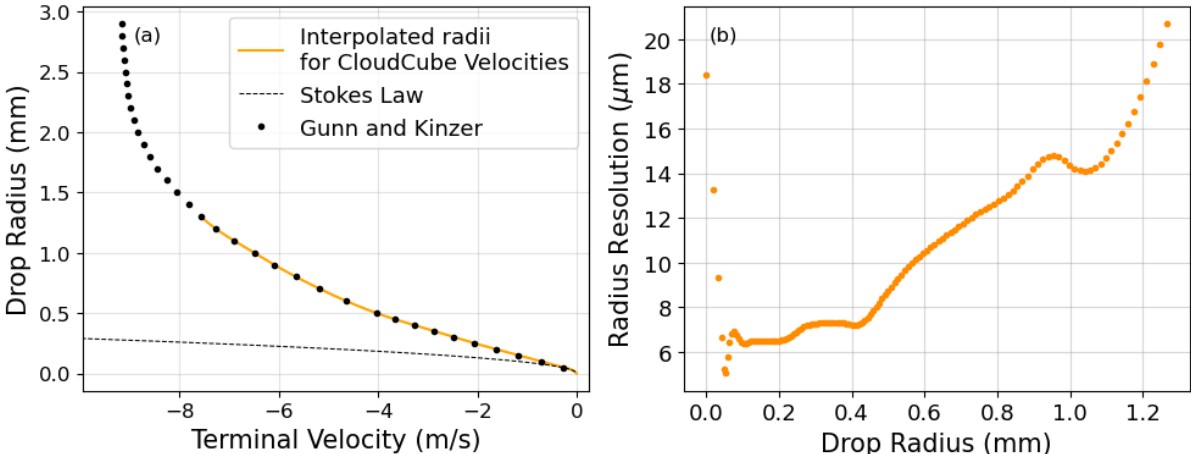

**Figure 4. (a)**: The terminal velocity relationships plotted along with interpolated radius values corresponding to the measured CloudCube spectrum velocities. The transition between the Stokes law regime and the Gunn-Kinzer interpolated points occurs at a radius of $50\mu m$. **(b)**: The radius resolution plotted as a function of the radii interpolated from the spectrum velocities.

spectra, we can assume that each collected Doppler spectrum, which are sampled once every 36 ms, is fully independent from the previous spectrum. Then, the spectrum error can be written as

$$\frac{\epsilon_s}{S} = \sqrt{\frac{1}{M}\left(1 + \frac{2}{\text{SNR}} + \frac{1}{\text{SNR}^2}\right)} \tag{4}$$

where $M$ is the number of averaged samples in our spectra and SNR is the signal to noise of each of the points in the spectrum. For CloudCube $M = 30$ spectra were averaged together before saving to disk. For attenuation, both water vapor attenuation and hydrometeor attenuation should be considered. Elevation dependent water vapor attenuation is derived using the temperature and relative humidity measured by the radiosondes (Rosenkranz, 1998). As shown in Fig. 5a, the hydrometeor attenuation is calculated and accumulated for each elevation where data to retrieve the DSD is available as per:

$$A_{tot}(R_N) = \sum_{n=0}^{N} \Delta A(R_n) \tag{5}$$

Attenuation accumulation is determined by assuming that the DSD stays constant between successive range bins. Then, the extinction coefficient can be assumed to be constant between $R(n)$ and $R(n+1)$, making the two-way optical depth $\tau = 2\int k(R)dR = 2k\Delta R$ (the factor of 2 accounts for the round trip distance made by a radar echo). To find $k$, the extinction coefficient, we need to integrate over extinction contributions from all particle sizes in the measured DSD: $k = \int k(r)dr = \int N(r)\sigma_{ext}(r)dr$. Here, $\sigma_{ext}(r)$ represents the extinction cross section of the particles, which is computed from the procedures described in Sec. 2.3. Then, the incremental attenuation contribution is determined by

$$\Delta A(R_n)\,(\text{in dB}) = 10\log_{10}(e^{-\tau}) = 10\log_{10}(e^{-2k\Delta R}) \tag{6}$$

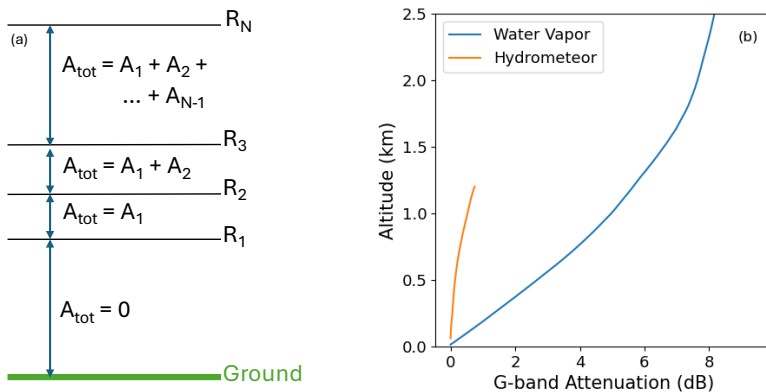

**Figure 5.** Left: G-band attenuation contributions by both water vapor and hydrometeors. Right: Diagram visualizing how hydrometeor attenuation is accumulated over elevation.

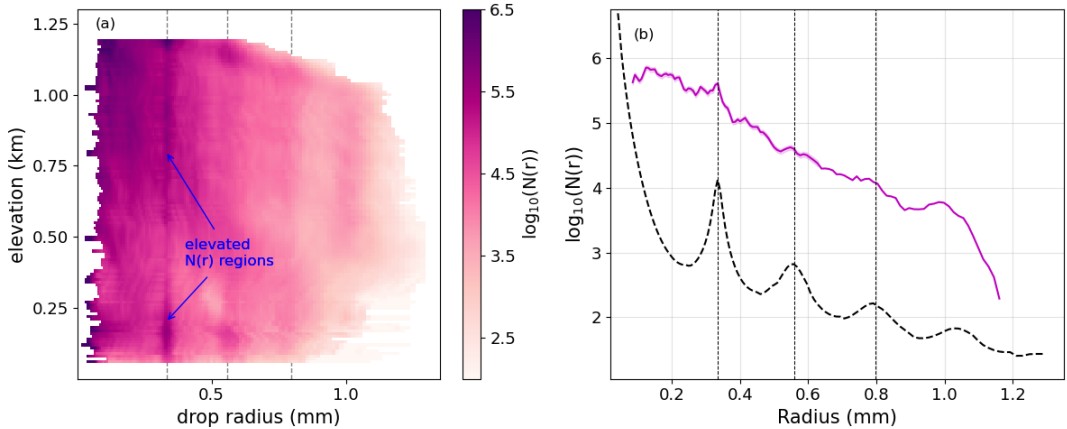

**Figure 6.** (**a**): DSD for each elevation calculated with the no turbulence assumption. Grey dotted lines represent particle radii that are at minima in the backscattering efficiency. (**b**): The pink curve shows the DSD at 1 km with the shaded region representing the $1\sigma$ instrument error, $\epsilon_s$. The vertical dotted lines represent the backscattering minima and the black dashed curve represents the limiting DSD values that CloudCube would have been able to detect at this elevation. CloudCube's sensitivity is -50 dBZ at 1 km. $N(r)$ is in units of $\text{m}^{-3}\,\text{m}^{-1}$

The relative importance of each of these attenuation contributions at G-band is shown in Fig. 5b. Because the precipitation
in this case is light, the attenuation is dominated by the water vapor. Fig. 6a shows the 2-D DSD retrieved from the example spectrum. There are lines of increased particle number density coinciding with the radii where backscattering minima occur. These are likely unwanted artifacts in the retrieval due to some combination of not taking into account turbulence, errors in the radius-terminal velocity relationship, or errors in the radius-backscattering efficiency relationship. To retrieve a 2-D DSD that mitigates these artifacts without a robust knowledge of the sources and magnitude of error, we implement a forward modeling
approach.

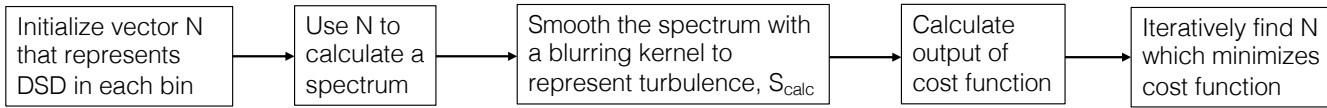

**Figure 7.** Block diagram explaining basic steps of forward model

## 5.2 Forward Modeling of Turbulent Spectrum

The forward modeling approach attempts to retrieves a vector that best represents the DSD. The model first uses an initial vector $N$ that represents the DSD to create an idealized spectrum using Equation 2. A log-linear best fit is used to the DSD calculated under the turbulence-free assumption to initialize. Then, this spectrum is smoothed with a blurring kernel that represents the effect of the turbulence to compute a spectrum that can be best compared with the measured spectrum. We minimize a loss function to find the most likely vector $N$. A diagram of this is depicted in Fig. 7.

Our forward model needs an estimate of the turbulence scale at every height. To do this, the framework described in O'Connor et al. (2005), which uses large scale turbulence to estimate smaller scale turbulence as such, is utilized:

$$\sigma_{v,turb}^2 = \sigma_{v,air}^2 \left( \frac{L_{small}^{2/3}}{L_{large}^{2/3} - L_{small}^{2/3}} \right) \tag{7}$$

The term $\sigma_{v,air}^2$ is the variance in the vertical wind speeds while the terms $L_{small}$ and $L_{large}$ represents the small and large length scales of the turbulence, respectively. The $L$ terms are dependent on the horizontal wind in the sight-line of the observation ($U$), the range of interest ($R$), the beamwidth of the radar ($\theta$), and the averaging time ($t$):

$$L = Ut + 2R\sin\left(\frac{\theta}{2}\right) \tag{8}$$

The short timescale, $t_{small}$ is the time between successive spectra (1.13 s for CloudCube). The long timescale $t_{large}$ is the total time over which the variance of the vertical winds are calculated. The variation of the vertical winds in the time vicinity of the example spectrum is shown in Fig. 8a. This figure also shows the horizontal wind speeds captured by the sondes at the same time as the example spectrum and the final turbulence derived from those values. Tthe turbulence scale is very similar to the velocity resolution of the spectra, so smoothing will have a relatively small effect on the DSD retrieval.

The most obvious choice of loss function for our forward model would be least squares. However, as we noted in the previous section, factors beyond turbulence correction are leading to unphysical artifacts in the retrieved DSDs. One way to retrieve a smooth DSD would be to impose a functional form for the DSD such as the modified gamma distribution (Deirmendjian, 1969). We take another approach. To encourage the retrieval of smoother and more physically realistic DSDs, we utilize a regularized least squares loss function:

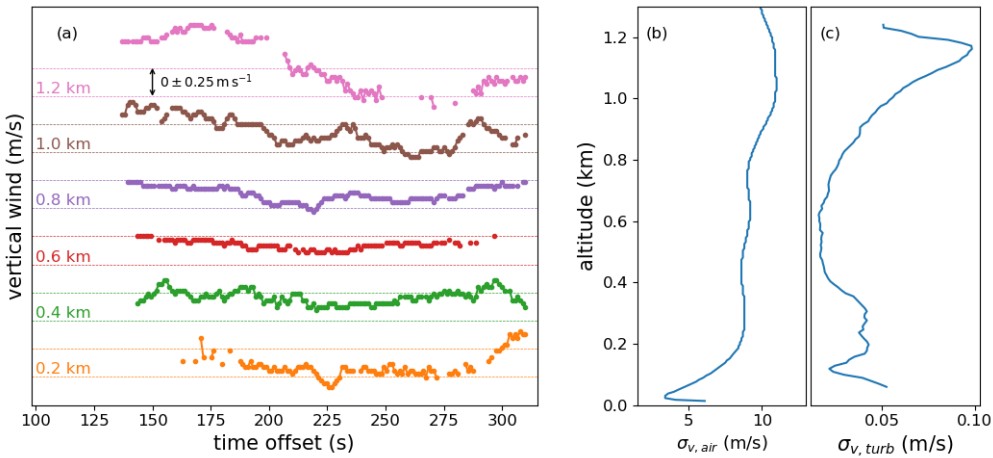

**Figure 8. (a)**: Depiction of how the vertical wind varies with time for a few different elevations. The dotted lines represent $\pm 0.25\,\mathrm{m\,s^{-1}}$ for each elevation. Plots of the **(b)** horizontal wind speeds measured by the sondes and **(c)** the final calculated turbulent broadening, in units of m/s

$$L = \sum \left[ \frac{(S_{calc,i} - S_{meas,i})^2}{\epsilon_{s,i}^2} + \alpha(N_{i+1} - N_i)^2 \right] \tag{9}$$

The first part of the loss function is a classic least squares loss. The second term represents the regularization. We use the total squared variation (TSV) regularizer, represented by $(N_{i+1} - N_i)^2$. This regularizer was first introduced by Kuramochi et al. (2018) as a way to enforce smoothness in 2-D imaging retrievals. The same principle applies to 1-D vectors, as penalizing the squared difference between adjacent points in a DSD favors a smoothly varying vector. The term $\alpha$ represents the regularizer weight, which determines how strictly we want to enforce vector smoothness. A large amount of regularization, meaning a

larger value for $\alpha$, will retrieve highly smoothed DSD vectors. We show a demonstration of this in Fig. 9.

We see that the pink curve (representing the highest amount of regularization we explored) produces a very smooth DSD and in turn a very smooth final spectrum. In Fig. 10, we show a 2-D DSD for the example spectrum, retrieved using the highest regularizer weight we tested, $10^4$. Compared to the turbulence-free DSD retrieval, we see a significant reduction in sharp gradients, though they are not completely eliminated. A primary issue with using regularized least squares, however, is that

we currently have no way to validate our choices of regularization. As the problem is over-constrained, different regularizer choices can yield final results with similar least squares errors. Without any ground truth data to train for the correct regularizer weight, we can only place confidence on the general shape and statistical properties derived from the DSD.

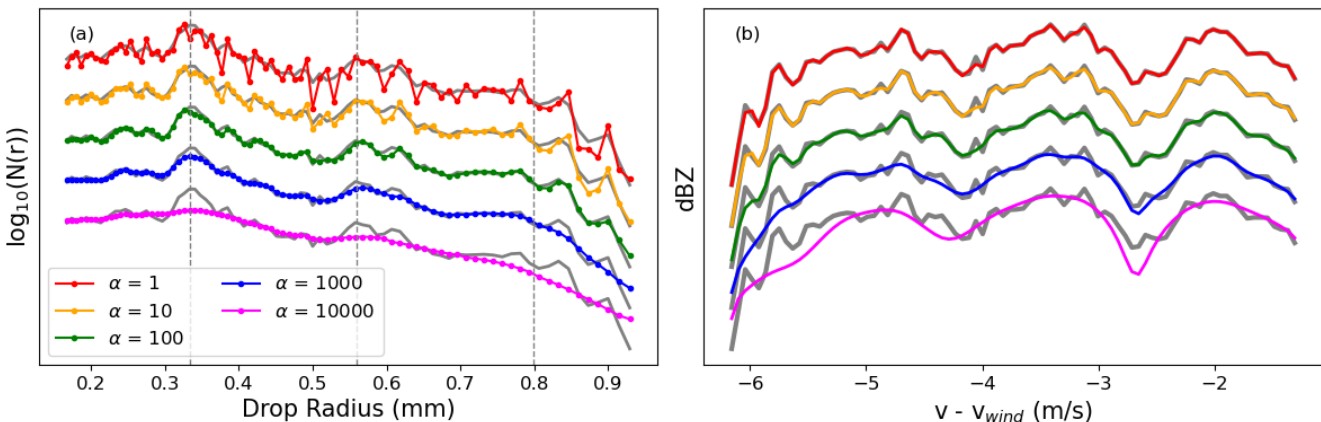

**Figure 9. (a)**: DSDs retrieved from the 1-D spectrum shown in Fig. 2d using 5 different regularizer weights. The grey curves plotted underneath the colored plots show the DSD computed from the turbulence-free assumption for comparison. The dotted vertical lines represent particle radii that are at minima in the backscattering efficiency. $N(r)$ is in units of m$^{-3}$ m$^{-1}$ **(b)**: The smoothed spectra derived from each of the DSDs. The grey curves underneath show the measured spectrum.

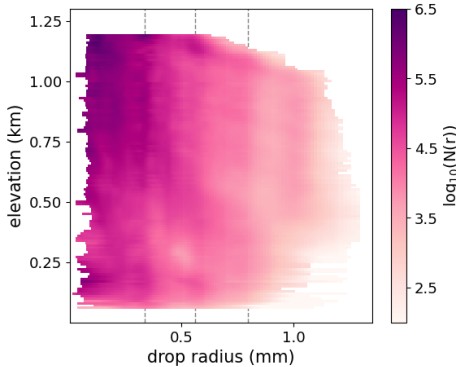

**Figure 10.** DSD for each elevation calculated using a regularized forward model and taking into account turbulent broadening. Grey dotted lines represent particle radii that are at minima in the backscattering efficiency. $N(r)$ is in units of m$^{-3}$ m$^{-1}$

## 6    Discussion

### 6.1    Sources of Uncertainty

Thus far in this work, we do not propagate errors to derive a final estimate of the uncertainty for the retrieved DSD values. While we acknowledge the importance of robust uncertainty analysis, many of the sources of error present are unable to be parameterized for easy propagation through all of steps of this retrieval. Therefore, this section will list each potential source of error and qualitatively comment on the sensitivity of the final DSD to this error.

The most significant source of uncertainty is the estimation of the vertical wind. Confidence in the vertical wind value
is dependent on the confidence of the minimum finding. The Gaussian mixture model used in minimum classification does provide an in-distribution probability for each classified point; however, there exists no clean way to relate this probability value to an uncertainty in m/s. Using minimum depth as a proxy for uncertainty also is not feasible, as there is no guarantee that the deepest minimum in a spectra actually corresponds to a backscattering notch. The terminal velocity is seen to affect the quiet-air estimate of the first minimum; however, several relationships are seen to provide a consistent theoretical first minimum
velocity. Hence, utilizing only the first minimum to compute vertical wind reduces uncertainty in the final wind estimate.

Offset in the radiosonde release time and the measurement time of the G-band spectra may introduce a small uncertainty as well. Important properties either directly measured by the sonde or computed from sonde measurements include air density, air temperature, gaseous attenuation, and horizontal wind. The largest source of uncertainty likely arises from temporal variation in the gaseous attenuation profile, which may fluctuate by up to a few dB if the humidity changes significantly. The horizontal
wind measurement directly affects the turbulent broadening calculation. For the case of EPCAPE, the turbulent broadening is minimal so temporal variations in the horizontal wind likely would not have a large affect. However, for measurements taken at much windier sites with large variations in the horizontal winds, a confident estimate on the turbulent broadening may be difficult to calculate. Air temperature and density are used to adjust the scattering functions and the terminal velocity functions respectively, though these only change by a few percent over the course of 1-2 km. Without a priori knowledge of the timescales
of the variability of these quantities, it is impossible to estimate an uncertainty value to be included in the retrieval process.

Another source of uncertainty that is challenging to quantify is the retrieval uncertainty. The uncertainty in the retrieved DSD introduced by regularization choices and initial condition choices cannot be parameterized. The measurement uncertainties for the spectrum data are known, but part of the retrieval uncertainty depends on the solution space for each model. The `scipy.optimize` function utilized in this retrieval provides an estimated covariance matrix; however, the high dimension-
ality of this problem likely leads to a complex solution space that is likely not explored thoroughly enough for good covariance estimates. Monte Carlo optimization techniques perform a more thorough search of the solution space generally, but such methods are too computationally expensive for high dimensional problems such as ours. Thus, a full error estimation for this forward model is highly non-trivial and is beyond the scope of this work.

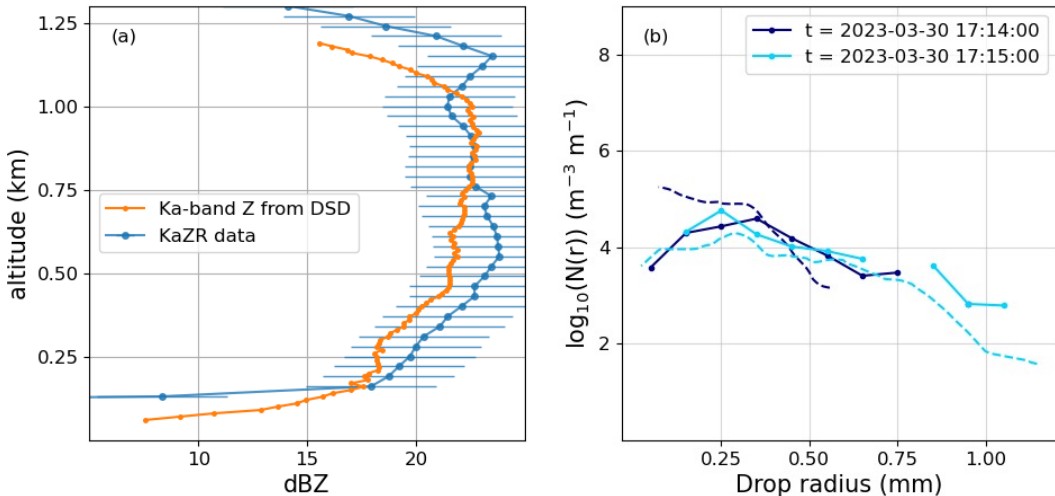

**Figure 11. (a)**: Ka-band reflectivity for each elevation calculated using the retrieved regularized DSD plotted with the Ka-band data collected by the ARM KaZR instrument. **(b)**: Comparison between the lowest elevation DSDs retrieved from the G-band spectra (dashed lines) and DSDs measured by the VDIS at the same time (solid line). The dark blue dashed line represents a measurement of the DSD at 200 m above the surface while the light blue dashed line represents a measurement 100 m above the surface.

## 6.2 Validation with Co-observing Instruments

Despite not fully understanding the uncertainty in the final retrieval, a first order understanding of the accuracy of the retrieved DSD can be found by utilizing data from both the ARM KaZR and VDIS instruments. Using the DSD depicted in Fig. 10 along with Equation 2, we can calculate a predicted Ka-band spectrum for each elevation. We use T-matrix scattering coefficients at Ka-band to calculate both the backscattering and extinction cross-sections. We correct for water vapor and hydrometeor attenuation in our theoretical spectrum before integrating across velocity to compute a single reflectivity value for each elevation.

This reflectivity can be directly compared to the reflectivity measured by KaZR, as shown in Fig. 11a. We can see that for the example spectrum, the predicted Ka-band reflectivity matches the KaZR fairly well, generally within a few dB. The presence of larger drops, which G-band instruments are not as sensitive to but Ka-band instruments are, may be affecting the accuracy of the predicted Ka-band reflectivity. Uncertainties in the hydrometeor attenuation also increase with elevation, potentially leading in higher inaccuracies in the Ka-band predictions as well. Still, the general consistency between the two curves gives

315 us some confidence in the quality of our retrievals.

We can use also direct measurements of the DSD taken by VDIS and compare it to the lowest elevation retrieved DSD we have available for the same time. The video disdrometer measures number densities in 0.1 mm radius increments, with a limiting drop size of 0.05 mm. However, the accuracy of the VDIS measurements below 0.1 mm is reduced due to the instrument struggling more to distinguish between smaller drop sizes. Additionally, the VDIS only saves data in 1 minute

increments, and unfortunately we only have a few minutes of data for which DSDs are able to be retrieved in the times

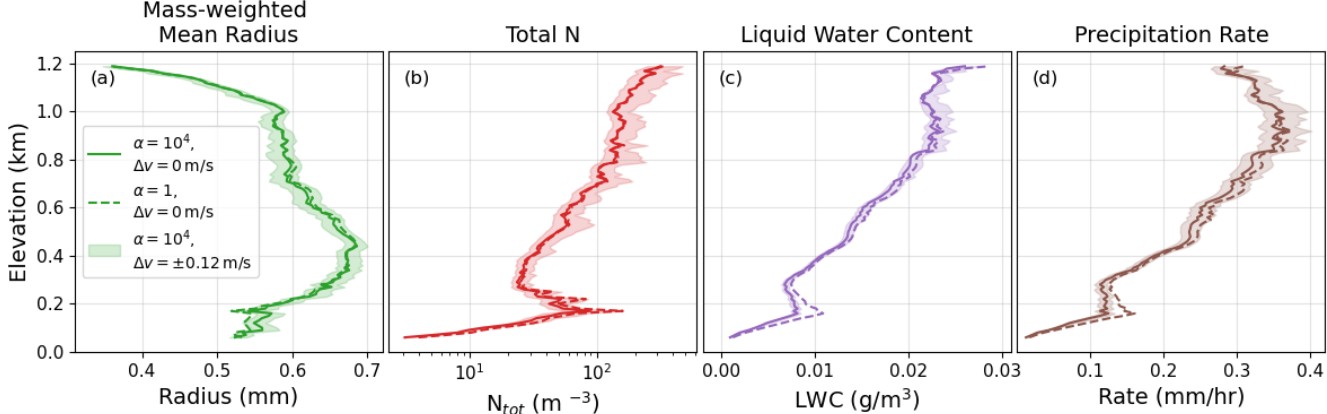

**Figure 12.** Visualization of estimation uncertainty of precipitation properties due to errors in the vertical wind retrieval as well as due to choices of regularization in the DSD retrieval. The solid lines represent parameters derived with the winds shown in Fig. 3 and with a regularizer weight of $\alpha = 10^4$. The dashed line shows properties derived with the same winds but with very little regularization, $\alpha = 1$. The shaded region represents properties retrieved with $\alpha = 10^4$, but assuming a $\pm 0.12\,\mathrm{m\,s^{-1}}$ deviation from the measured vertical wind. This represents two spectrum bins away from the best estimate.

adjacent to the example spectrum. Thus, there are only two coincident times between the CloudCube measurements and the VDIS measurements, plotted in Fig. 11. We see similarity between the retrieved DSD and the VDIS measured DSD, with discrepancies being the highest at the smallest drop sizes. Because of the significant fall time of the small droplets from the lowest DSD elevation (typically around 50 m) to the ground, comparing measurements with the same time stamps is perhaps
comparing slightly different drop populations. However, with a very limited amount of G-band data of good enough quality to retrieve DSDs (typically only on the order of a few minutes), and the slow sampling time of the VDIS, it is challenging to compare measurements with a sufficient lag time to account for the fall time.

## 6.3 Estimating Bulk Precipitation Properties

Despite the various uncertainties described above, we can derive bulk properties of the DSDs, which are easier to use and
330 should be more robust to uncertainties. Here we derive four bulk properties of the distribution. Fig. 12 shows the plots of mass-weighted mean radius, total number density, liquid water content, and precipitation rate derived from the DSD of the example spectrum. Mass-weighted mean radius, $R_m$, is calculated as

$$R_m = \frac{\int N(R)M(R)R\,dR}{\int N(R)M(R)\,dR} \tag{10}$$

where $M(R) = \rho_w V(R) = \rho_w (4/3)R^3$ is the mass of a water droplet with radius $R$, where $\rho_w$ is the density of liquid water
and $V(R)$ is the volume of a drop with radius $R$. The total number density is simply calculated as $N_{tot} = \int N(R)\,dR$, the liquid

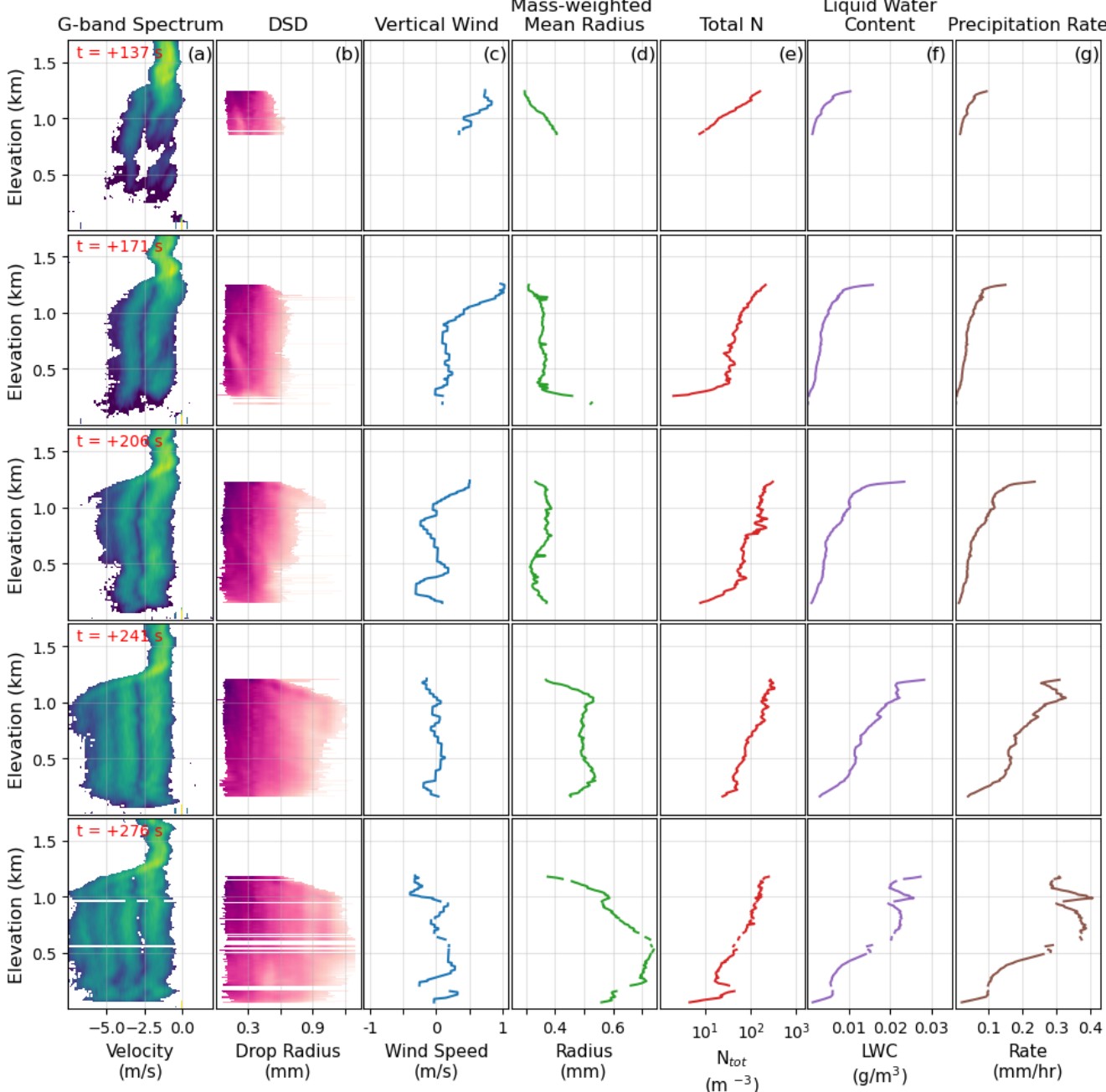

**Figure 13.** Compilation of statistics possible to be calculated from DSDs retrieved shown for 5 G-band spectra spaced 35 seconds. Times shown in left column are seconds elapsed since 2023-03-30 17:10:18 UTC. Gaps in the DSD are due to minima corresponding to the first backscattering minimum not being available at that elevation. Especially in the bottom row, this can be seen for elevations where the minimum is actually below the noise floor and is therefore not detected by our algorithm. Spectra are shown in dBZ, using the same colorbar as used previously. $N(r)$ is in units of m$^{-3}$ m$^{-1}$, using the same colorbar as used previously.

water content is calculated as $LWC = \int N(R)M(R)\,dR$, and the precipitation rate is calculated as $P = \int N(R)V(R)v(r)\,dR$. Profiles of these properties are shown for the lowest and highest explored values of the regularization weight, $\alpha$, as well as for $\pm 0.12\,\mathrm{m\,s^{-1}}$ err in the vertical wind speed (representing $\pm$two bins in the measured Doppler spectrum). We see that while these errors may affect the details of the binned DSDs, the bulk precipitation properties are relatively insensitive to choice of
regularization weight and fairly robust to vertical air motion uncertainties.

Figure 13 shows the plots of these precipitation properties derived from the DSDs of five different spectra, spaced 35 seconds apart. This figure highlights the rapid timescale of variability present in these drizzling systems. Coarse sampling times in measurements of the precipitation properties are at risk for missing important details in the cloud and precipitation processes.

## 7   Conclusions

We have presented a retrieval methodology to derive the vertical wind and the precipitation DSD in light rainfall from a nadir pointing G-band Doppler spectrum. This work extends the methods developed for W-band to lighter rainfall than has been possible to date. The G-band retrievals work well for light precipitation because the first Mie notch occurs near a radius 334 microns, thereby enabling accurate estimation of the wind speed for very light precipitation rates. Furthermore the precipitation water contents are very small so the attenuation from condensed water is insignificant relative to the gaseous attenuation.
As pointed out by Courtier et al. (2024), the method demonstrated here would optimally be combined with multi-frequency W- and K-band Doppler spectra (e.g. Tridon and Battaglia, 2015) to seamlessly extend from the lightest to heavy precipitation events.

There are residual uncertainties in the binned DSD due to inaccuracies in the droplet fall velocity and drop obliquity relationships, which appear as ripples near the location in the spectrum where Mie notches are present. We expect some uncertainty
contribution from factors such as errors in the minimum finding routine, errors in estimation of turbulent broadening effects, errors in computed gaseous attenuation, and discrepencies in air temperture and density which propagate into errors in the DSD retrieval. Nevertheless, the bulk statistics of the DSD, such as the water content, number concentration, precipitation rate, and mass weighted mean size are relatively robustly derived. We note that the retrievals presented here are for high-SNR cases with clearly defined notches in the spectrum. Future works will refine the algorithm presented here to work for spectra with
less cleanly defined spectral features.

With the growing number of G-band radar observations (including CloudCube G-band's ongoing participation in the Cloud And Precipitation Experiment at kennaook; Mace et al., 2023) the Doppler-spectral retrieval method offers the unique potential to provide profiles of light rainfall and drizzle in stratocumulus and shallow cumulus clouds relative to approaches centered on the radar reflectivity.

*Data availability.*   TEXT

All datasets can be found at the following references. CloudCube Doppler Spectra: https://doi.org/10.5281/zenodo.11043877 (Socuellamos et al., 2024b), KaZR: https://doi.org/10.5439/1498936 (Lindenmaier et al.), VDIS: https://doi.org/10.5439/1992988 (Zhu and Wang), Ceilometer: https://doi.org/10.5439/1181954 (Zhang et al.), Radiosonde: https://doi.org/10.5439/1595321 (Keeler et al.)

*Author contributions.* TEXT

MDL coordinated the participation in EPCAPE. RRM, KBC and JMS built CloudCube's G-band radar. JMS processed the CloudCube data sets and provided guidance on their utilization. NYY and MDL developed methodology and codes for the presented retrievals. NYY composed the manuscript in collaboration with the rest of the authors

*Competing interests.* TEXT

At least one of the (co-)authors is a member of the editorial board of *Atmospheric Measurement Techniques.*

*Acknowledgements.* This work was performed at the Jet Propulsion Laboratory, California Institute of Technology, under a contract with the National Aeronautics and Space Administration. KAZR, ceilometer, disdrometer, and radiosonde data were obtained from the Atmospheric Radiation Measurement (ARM) user facility, a US Department of Energy (DOE) Office of Science user facility managed by the Biological and Environmental Research Program. Nitika Y. Yurk's research was supported by an appointment to the NASA Postdoctoral Program at the
380 NASA Jet Propulsion Laboratory, administered by Oak Ridge Associated Universities under contract with NASA. The authors thank Zeen Zhu for insightful information regarding the ARM disdrometer instruments.

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
