# Peer review of "Vertical Wind and Drop Size Distribution Retrieval with the CloudCube G-band Doppler Radar"

_EGUsphere, 2025_

## Referee Comment (RC2)

[revised manuscript text omitted]

filtered out. Our final check is to ensure that we are not erroneously finding points in the noise floor of the spectrum as minima by ensuring that for true detections there is continuous data for at least $1\,\mathrm{m\,s^{-1}}$ on each side of the minimum. A demonstration
125 of this minima retrieval for a single elevation is shown in Fig. 2c. The spectrum in Fig. 2b will be referred to hereafter as "the example spectrum" and will be used to demonstrate our methods for the remainder of this paper. To provide context for this example spectrum, Fig. 2a shows the G-band reflectivity curtain and mark the time at which the example spectrum was collected. Also shown is the cloud base height as measured by the ARM infrared laser ceilometer (Morris, 2016). This ceilometer works best in non-precipitating conditions and the presence of drizzle is likely responsible for the sharp variations
130 in observed base height.

[revised manuscript text omitted]

**Figure 9. (a)**: DSDs retrieved a single elevation from the example 2-D spectrum using 5 different regularizer weights. The grey curves plotted underneath the colored plots show the DSD computed from the turbulence-free assumption for comparison. The dotted vertical lines represent particle radii that are at minima in the backscattering efficiency. **(b)**: The smoothed spectra derived from each of the DSDs. The grey curves underneath show the measured spectrum. The errors shown here are derived using the covariance matrix produced from the fit. For now we only consider the variance of individual point and ignore correlations between neighboring points.

[revised manuscript text omitted]

  With the growing number of G-band radar observations (including CloudCube G-band's ongoing participation in the Cloud And Precipitation Experiment at kennaook) the Doppler-spectral retrieval method offers the potential to provide unprecedented observations of profiles of light rainfall and drizzle in stratocumulus and shallow cumulus clouds relative to approaches centered

310 on the radar reflectivity.

*Data availability.*

  The CloudCube G-band Doppler spectra described in this article are provided in netCDF format in the file titled Cloud-Cube_EPCAPE_Gband_Spectra.zip at https://doi.org/10.5281/zenodo.10076227 (Socuellamos et al., 2024b).

  The data captured by ARM instruments (KaZR, VDIS, laser ceilometer) that were used in this article can be found at the

315 following link: https://www.arm.gov/research/campaigns/amf2023epcape

[Figure]

[Figure]

*Author contributions.*

[revised manuscript text omitted]

---

## Author Response (AR1)

*We thank the reviewers for such a detailed read of our manuscript and for insightful comments to improve the clarity and impact of the paper. We have made several significant changes to the manuscript including adding an extended qualitative discussion of the uncertainties in the retrieval, adding more detail regarding the context of the measurements we use to demonstrate our method, and clarifying our retrieval process in places where the descriptions were insufficient. We provide a point-by-point response to all of the reviewers' comments below. The reviewers' comments are in regular font followed by our responses are in bold font. We believe that these modifications satisfy the reviewers initial concerns.*

**Response to Reviewer #1**

Major comments:

- Methodology: How is the minima depth dependence on the noise floor (l. 119-120) justified? For example, why should we trust more, or better say choose a much deeper minimum in case of a higher noise floor? If the minimum is well above or even slightly above the noise floor, why does it matter? If that is the actual algorithm implementation, then we potentially introduce a significant caveat here. There could be other methods to define a the notches such as a certain reduction in minima's signal amplitude relative to the closest peaks, but my understanding is that this is not the case here.

  **Our description of the procedure appears to be unclear. We do not rely on the minimum being higher than the noise floor. We require that the depth of the minimum (the peak to trough distance) be at least 5x the amount of random fluctuation we expect from Rayleigh distributed noise. This ensures we have not picked up a large fluctuation as a minimum. The language in lines 119-120 is amended to be clearer**

- Methodology: Given that the raindrops follow some PSD, which is then manifested in the resolved spectra, there can be some offset in the notch location. How big is this offset? Have you examined how the retrieved PSD (in the following steps) propagates into the exact notch location or whether using different Gamma PSD input parameters modifies the exact notch location? I suspect this offset will be smaller than the mode differences illustrated in fig. 3c, yet this should be accounted for, or at least acknowledged as an

uncertainty source (e.g., could this be a source for the minima inconsistencies discussed in l. 164-166?)

*We investigated this effect initially and found that for a gamma PSD, changing the parameters of the model does not affect the location of the minima in the Doppler spectrum. See the plot below where we investigate several different model parameters, using models from two different papers and trying two different terminal velocity relationships (dashed vs solid line). While the details of the terminal velocity relationship can significantly affect the location of the minima, in general for a monotonically decreasing PSD we do not see an impact in the minima locations. We add a sentence in this paragraph near l. 166 discussing this.*

[Figure]

- Methodology: turbulent broadening calculations (l. 223-234): I could be wrong here but it seems to me that this paragraph has many inaccuracies, e.g., how was eq. 7 derived (this exact format is not in O'Connor et al. (2010)? Where are t_small and t_large referred to in the equations? L in eq. 8 only refers to the large scale (scattering volume) horizontal dimension, correct? Then what is the definition of Lsmall? I presume that t_small is simply the radar averaging time often equivalent to the dwell time, is that correct?

  Perhaps most importantly, how is the dissipation rate retrieved (implicit in sigma_v_air)? This could have a critical impact on the forward-calculated variables and the conditions discussed in O'Connor et al. (2010) might bot be applicable here, for example

  Are those inaccuracies mentioned above implemented in the actual retrieval?

How is the resolved turbulent broadening convolved in the actual spectra as per eq. 1?

***Thank you for noticing these inconsistencies. The root cause of this is that we reference the wrong O'Connor paper here. We meant to reference O'Connor 2005. The reference is updated now.***

- Uncertainty quantification: given all the "moving parts" in this retrieval, I find it hard to believe such small uncertainties as the authors present and discuss (e.g., fig. 12 l. 288-290) are representative, especially given that the depicted uncertainties are merely the propagation of an arbitrary value (two bins) and nothing else. The uncertainty quantification should be revisited here and potentially discussed in more detail (e.g., could the second and third minima be incorporated, when detectable, as an uncertainty metric, e.g., as illustrated in fig. 3c? could the confidence in the Gaussian mixture model output be incorporated (or not - a possibility)? What about the turbulent broadening estimates? Does the time since radiosonde release impacts the uncertainty? etc.

  ***We added another subsection in Section 6 which discusses errors and their potential effects on in more detail. Here our intention is to point out to the reader all of the potential sources of uncertainty. However, we do not believe a thorough quantitative accounting of all errors is achievable for this paper as it requires a substantial amount of analysis. We believe this may be more appropriate for a future work, where our retrieval algorithm is more refined.***

- Text consistency: using radii instead of diameters is somewhat confusing, potentially deceptive, and sometimes used interchangeably. l. 61 provides an example where such a radii-diameter confusion comes into play. Drizzle is defined as drops with diameters smaller than 0.5 mm, not radii, so from this definition, even the G-band notch method applies to rain drops, not drizzle (by the self-definition of rain drop). Given that ostensibly drop diameters are more commonly used in the community and literature, I recommend converting all drop dimensions in the text and figures to diameters and check such claims as in l. 61.

***We agree this can be confusing however we find it necessary to retain both radius and diameter while clearly noting which measure of size we use in each location. Specifically, the equation governing the Doppler spectrum and DSD relationship we rely on for this analysis (from Kollias 2011) is in terms of***

*radius, which is why we chose to do our analysis in terms of radius. We think to be consistent with that paper, we prefer to keep our analysis in terms of terms of radius. On the other hand the parameterization of aspect ratio found in Thurai and Bringi (2005) is given in terms of diameter. We feel it is best to keep these equations in the form of the original citations.*

*We do fix the definition of drizzle to match what is in the AMS glossary.*

Minor comments:

- Abstract l. 15 - "These bulk properties are relatively invariant to the assumptions made in the estimation of the full DSD retrieval..." - this claim is unsupported by the text and I doubt it can be supported using a single case study without a rigorous evaluation. Either rework the DSD impact on *all* retrieval components or omit this part of the sentence, e.g., start the sentence with "We suggest that large volumes of such retrievals can be useful tools ..."

  *We agree that we should not claim that this will be true for all cases. We edit the end of abstract to clarify that we find this to be true for our case study only include that further validation of this method would be useful for substantiating these claims*

- l 27 - observations --> measurements

  *We have revised this as suggested*

- l 41 - Dopper moments --> higher Doppler moments (since reflectivity is the 0th moment)

  *Agreed, we have revised this as suggested*

- l 85 - refer to the fact that EPCAPE was a DOE ARM campaign. This is noted below on a different form but should be stated at the beginning. Also, add a

reference for EPCAPE (likely the science plan -
https://doi.org/10.2172/1804710)

***Thank you, we add the references noted as suggested***

- l 88 - This should come in the first sentence of this paragraph - also define the ARM acronym

***We ensure the first sentence of the paragraph contains the relevant acronyms***

- l 90 - data iteself is --> datasets are (data is plural of datum – change here and elsewhere where applicable, e.g., l. 141)

***We agree and have made this edit throughout the text***

- l 95 - refer to the ARM radiosonde handbook - https://doi.org/10.2172/1020712.

***Thank you for providing the link to the handbook. We add the reference***

- l 97 – Add a reference to the KAZR handbook - https://doi.org/10.2172/1035855

***Thank you for providing the link to the handbook. We add the reference***

- l 99 - that is the wrong reference (LD handbook) - change to VDIS handbook - https://doi.org/10.2172/1251384 - note that the title on osti is wrong but the document is correct).

***Thank you for catching this error. We correct the reference***

- l 108-110 - What is "sufficient span of elevations and times"? This should be elaborated as per the first paragraph of my review.

***We add a quantitative description of what we mean by "sufficient" in the beginning of the paragraph. Additionally, we add a sentence emphasizing that these values were chosen arbitrarily and were not optimized in any specific way.***

- l 108 - campaign --> deployment

***We have edited the wording as suggested***

- l 117 - Add reference to scipy

***Thank you for this reminder, we have added the reference***

- l 125-130 – The example case should be described in detail, and a panel showing the mean Doppler velocity should be added to Fig. 2. For non-experts, the

melting layer echoes are not clear, and questions could arise concerning the cloud base height vs. melting layer indications. Without further elaborating on this scenario, one might think that we have a ringing effect rather than big ice particles melting and generating the interesting observed spectra consisting of a wide range of rain drop sizes and therefore large spectral widths, evolving *below cloud base*.

*We agree that a more detailed explanation of the example spectrum strengthens this section. We add a mean doppler plot as panel (b) of Figure 2 and include an extra paragraph in the opening of Section 3 introducing the data further.*

- l 128-130 – quick-looking at other ARM data from this case, those ceilometer variations looks physical to me. Recommend removing this sentence.

*Thank you for this advice, we remove the sentence*

- l 131 - by "single 2-D spectrum" do you refer to a spectrogram (Doppler array values vs. height) or something else?

*We agree this was unclear, we clarify that we mean a Doppler profile from a single time*

- l 138 - provide reference to SciKit-Learn

*Thank you for the reminder, we add the reference*

- l 147-149 - the stark outlier is discussed but how is the cyan pint at ~250 m next to the purple points justified?

*We add a few sentences at the end of this paragraph explaining further our outlier identification method and the shortcomings of it that may lead to points such as the one you point out not being flagged as outliers.*

- l 159 - define vt and k

*We clarify that k is a constant in this equation and add the definition of vt*

- l 159-160 - that is incorrect. Air density is not measured by the radiosonde, but can be calculated using radiosonde measurements.

*Thank you for catching this, we correct this in the text*

- l 178 - |K^2| (the dielectric factor) is not the square magnitude of the refraction index, but it depends on it

  *Thank you for correcting this, we made changed the wording*

- l 183-184 - "In the Stokes regime, …" - this sentence is unclear - recommend rewording.

  *We agree this is unclear, we change the text to just refer to regimes separated by specific radius thresholds*

- l 185-186 - can you explain the local minima in fig. 4b?

  *We believe these are just due to measurement errors in the Gunn-Kinzer points. We add a sentence at the end of this paragraph clarifying this*

- l 195 - what does "previous" refer to?

  *We mean "previous spectrum," – we fix the wording in this sentence to reflect that.*

- l 204 - this is the two-way optical depth so either define it as such or remove the factor of 2 here and add it in eq. 6.

  *We see the confusion we created here – we clarify "two-way" in the text.*

- l 244 - recommend using an alternative nomenclature for lambda because it is immediately associated with the wavelength rather than the regularization weight.

  *We agree, all instances have been changed from lambda → alpha*

- l 228 - t is the radar averaging time, not the turbulence time scale

  *We agree, we have corrected this*

- l 271-277 - I am not sure I'd consider the depicted PSDs as showing good agreement. The reason here is that one would expect the drops to evaporate between the lowest (out-of-cloud) range gate and the surface; hence, one would expect a PSD offset to the left (decreasing sizes) in the case of the VDIS, whereas in the plotted spectra (fig 11b), the shift is to the right (apparently increasing sizes). I recommend (a) revisiting the text here and (b) I am less familiar with the CloudCube radar and do not know how susceptible it is for near-surface clutter effects, but it might be good to also examine the retrieval 50 or 100 m above the first range to reduce that probability - even if we are

further away from the surface, the signal might be cleaner and hence, might generate better agreement with surface observations.

*We reword this statement to just point out the similarity between the two DSDs rather than stating that they are consistent with each other.*

- l 307 - provide reference for that deployment

    *We add a reference to the published science plan*

- l 285 - define rho_w

- l 287 - define the volume in the equation

    *We add the definitions of both of these terms in the sentence referred to*

- l 308 - "to provide unprecedented observations of profiles of ..." such microphysical quantity retrievals are not unprecedented. Perhaps you meant something like: "to robustly provide profiles of ..."?

    *We reword just to highlight that this instrument is unique in being able to provide profiles in drizzling conditions*

- Data availability statement: The KAZR and VDIS datasets require references including their DOI, which I believe can be retrieved from ARM.

    *We add references to each of these datasets as recommended*

- Table 1 - define FMCW. Also, many of those parameters (at least for KAZR) are configurable so the caption should specifically refer to EPCAPE.

    *Definition added in caption*

- fig 3 onward - provide the units for each plotted spectrograph as in fig. 2b

    *We add a colorbar with units specific in all spectra in the paper, except for the spectra shown in Figure 13. We feel that this figure it quite large as is and adding a colorbar would take away from the readability of the figure*

- fig 6 and elsewhere - add units to for N(r) within or next to the logarithm.'

    *We add the units for N( r ) in the captions for figures where DSDs are shown.*

- fig 8 - turbulence velocity? Do you mean turbulent broadening?

  *Yes, we correct this wording*

- fig 9: caption - this seems like a spectrum (1D) rather than 2D. To which example spectrum (altitude and time) do you refer here?

  Also, is the reflectivity here really in linear units? This appears inconsistent with the dBZ units in fig. 2b. Same goes for N(r), which seems more log-scale than linear.

  *We revise these plots to be in log scale so that they are consistent with the rest of the plots in the paper. Also add clarification for which 1-D spectrum the retrievals are done at*

- fig 10 - specify RML

  *Actually we don't need this acronym and just replace it with the word "regularized"*

- fig 11 - nice results - I'd consider 2 dBZ difference (at least up to 1.1 km) to be well within the KAZR measurement uncertainty (3 dBZ; see handbook). To support your case, adding such error bars/uncertainty envelope around the measured profile would be helpful to demonstrate that the forward calculated values are within the uncertainty range.

  *Thank you! We agree with this suggestion and add errorbars to the figure*

- fig 13 - specify the times each spectrograph corresponds to.

  Also, I think that a large fraction of readers would be interested in the vertical air motion aspect of this retrieval (as the manuscript title suggests) - recommend adding profiles to this figure.

  *We agree that vertical winds should be highlighted here and add a panel to show the winds. We also add some text in the caption to clarify what the time stamps in the figure stand for.*

***Response to Reviewer #2***

**Line 15-16 (Page 1):** This feels like too much, given the level of discussion provided on these properties. Either remove from the abstract or provide more evidence

***We agree and this is consistent with comments received from reviewer 1. The end of the abstract is modified to limit our conclusions***

**Line 17 (Page 1):** You haven't mentioned it as far as I can see, so maybe a quick discussion of why you're only analyzing the liquid portion of the cloud when you do also observe the ice is appropriate

***We add a few sentences to the end of the second paragraph of section 3 discussing why we do not perform any analysis of the ice particles.***

**Line 28 (Page 2):** aerosol-cloud

**Line 28 (Page 2):** cloud-climate

***We add the dashes to both of these word pairs***

**Line 70 (Page 3):** Presumably "liquid water"?

**Line 70 (Page 3):** "precipitation" not necessary, attenuation doesn't mind if liquid water is precipitating or not

***We rephrase this sentence on line 70 to be more clear, addressing both of the above comments***

**Line 71-72 (Page 3):** What do you mean by this? The frequency isn't what you have referred to above. Indeed the study mentioned talks about the extra care needed in attenuation correction at smaller wavelengths.

***We rephrase this to clarify that we mean our attenuation correct errors are lower than errors seen by Ka- and W-band radars in conditions with more attenuation present.***

**Figure 1 (Page 3):** Typically the drop diameter is used rather than radius (I suspect that this is the answer to my comment about the drop obliquity later). Please change figures and references in text from drop radius to drop diameter

*We do acknowledge that in many other texts, drop diameter is typically used. However, the equation governing the Doppler spectrum and DSD relationship we rely on for this analysis (from Kollias 2011) is in terms of radius, which is why we chose to do our analysis in terms of radius. We feel that to maintain consistency with that text, it would be best for us to keep our analysis in terms of radius.*

**Line 95-96 (Page 4):** Difficult to follow this, please rewrite

*We agree the original wording was a bit confusing, we rewrite it for clarity.*

**Section 2.2 (Page 4):** Why are the W-band and Ka-band not used in the validation?

*Firstly, Doppler spectra were not available at these frequencies. Secondly, the calibration of the ARM KaZR radar was of better quality than the Ka-band CloudCube radar, thus was preferred for validation. An ARM W-band radar was not available at this site for comparison as far as we could tell.*

**Line 103-104 (Page 4):** If the aspect ratio of the droplets is 1 at r<0.84mm why do you only trust the first Mie notch to be unaffected by the obliquity parameterisation?

*The referenced paper is in terms of diameter so this value 0.84mm is also a diameter – we add text to clarify*

**Line 164-169 (Page 8):** Would you trust the wind speed from the first minimum even if the other two agree on a different speed. E.g. at ~600m in fig. 3c where the second and third minima say -0.1 m/s and the first minimum says 0.1 m/s?

*We agree that understanding the uncertainty around the derived vertical velocity is challenging to quantify. We discuss this qualitatively in more detail in the newly added Subsection 6.1*

**Line 249 (Page 12):** What level of regularization is used?

*We include the regularization level in the text in this sentence*

**Line 249 (Page 13):** What is RML?

*We remove this and just replace with the word "regularized"*

**Line 248-249 (Page 13):** A comparison to the measured spectrum would seem appropriate at least

*This is part of the core issue with regularization - many different regularizer choices may give fairly similar least squares errors with the measured spectrum. We add a sentence about this here.*

**Line 267 (Page 14):** What elevation is this? presumably this has been chosen to avoid any near-surface effects

*We add the elevation information into the caption of Figure 11*

**Line 270-274 (Page 15):** I agree that the 17:15 comparison is a good match. I think the 17:14 perhaps is less so, especially at small sizes given the log scale. Could you comment on the large difference in DSD in consecutive minutes for the G-band, especially given the consistency in the disdrometer

*We suspect this may just be due to rapid changes in the DSD itself in the "blind zone" we have between the lowest elevation we can see a DSD for and the surface, as we allude to at the end of this paragraph. But we do not wish to make any conclusive statements about it in the text.*

**Line 276-277 (Page 15):** It appears from FIg 2 that you have enough data to do at least one minute long integration of the retrieved DSD to compare to the disdrometer DSD in a more apples-apples way

*We considered this, but felt that given the other limitations of this method, it would still not be a strong point of comparison. We just aim to show general consistency with the VDIS measurements but not place a lot of emphasis on it as a robust method of validation.*

**Line 295-297 (Page 16):** If the first notch is not detected, but subsequent notches are, why can these not be used as an estimate of the vertical wind?

*They certainly could! For a future work we are exploring using non-masked data for the retrieval to address this problem.*

**Minor Editorial Comments (Page 8):**

- Insert comma after "regime"
- Insert comma after "decreases"
- Change "again increases" to "increases again"

*We fix these minor changes as suggested*